# Isolation and Characterization of a ssDNA Aptamer against Major Soluble Antigen of *Renibacterium salmoninarum*

**DOI:** 10.3390/molecules27061853

**Published:** 2022-03-12

**Authors:** Brady Layman, Brian Mandella, Jessica Carter, Haley Breen, John Rinehart, Anna Cavinato

**Affiliations:** 1Department of Chemistry and Biochemistry, Eastern Oregon University, La Grande, OR 97850, USA; blayman@eou.edu (B.L.); bmandella@eou.edu (B.M.); jcarter@eou.edu (J.C.); hbreen@eou.edu (H.B.); 2Department of Biology, Eastern Oregon University, La Grande, OR 97850, USA; jrinehar@eou.edu

**Keywords:** bacterial kidney disease, aptamer, SELEX, GO-SELEX

## Abstract

Bacterial kidney disease (BKD) is a major health problem of salmonids, affecting both wild and cultured salmon. The disease is caused by *Renibacterium salmoninarum* (Rs), a fastidious, slow-growing and strongly Gram-positive diplobacillus that produces chronic, systemic infection characterized by granulomatous lesions in the kidney and other organs, often resulting in death. Fast detection of the pathogen is important to limit the spread of the disease, particularly in hatcheries or aquaculture facilities. Aptamers are increasingly replacing conventional antibodies as platforms for the development of rapid diagnostic tools. In this work, we describe the first instance of isolating and characterizing a ssDNA aptamer that binds with high affinity to p57 or major soluble antigen (MSA), the principal antigen found on the cell wall surface of Rs. Specifically, in this study a construct of the full-length protein containing a DNA binding domain (MSA-R2c) was utilized as target. Aptamers were isolated from a pool of random sequences using GO-SELEX (graphene oxide-systematic evolution of ligands by exponential enrichment) protocol. The selection generated multiple aptamers with conserved motifs in the random region. One aptamer with high frequency of occurrence in different clones was characterized and found to display a strong binding affinity to MSA-R2c with a *K_d_* of 3.0 ± 0.6 nM. The aptamer could be potentially utilized for the future development of a sensor for rapid and onsite detection of Rs in water or in infected salmonids, replacing time-consuming and costly lab analyses.

## 1. Introduction

Renibacterium salmoninarum (Rs) is the causative agent for bacterial kidney disease (BKD) which affects wild and cultured salmonids and has a serious impact on the salmonid industry worldwide [1]. Prevention and control of Rs infections are difficult because there is no reliable vaccine and certain Rs strains can develop antibiotic resistance. Traditional methods to detect Rs are based on bacterial culture of infected tissues. While this method can be very sensitive, it is time consuming due to the very slow growth of the bacterium. Immunological and molecular methods are available including the fluorescent antibody test (FAT), enzyme-linked immunosorbent assay (ELISA), and non-quantitative and quantitative procedures for the polymerase chain reaction (PCR) [2]. ELISA is the most widely used method, due to the low cost and ability to rapidly screen large numbers of fish. The principal protein detected by ELISA is an abundant 57 kDa protein called p57 or major soluble antigen (MSA), present on the cell surface wall of Rs [3]. The putative sensitivity of ELISA has been estimated to be several hundred cells per reaction [4] or approximately 9 × 10^6^ cells/g of tissue [5]. However, these methods focus primarily on tissues such as kidney or ovarian fluid, thus requiring that fish be sacrificed and are not suitable for onsite detection of the bacterium.

Aptamers are short, single-stranded sequences of nucleic acids that can be selected in vitro from randomized DNA or RNA synthetic libraries using SELEX (systematic evolution of ligands by exponential amplification). DNA aptamers are stable and thus ideal for sensor applications even in harsh conditions. They display very high target specificity and superior binding affinity for the target molecule with dissociation constants from nano to picomolar range and other features that make them more appealing than antibodies [6]. Since first reported in the 1990s, several SELEX methods have been developed to facilitate more efficient isolation of aptamers for a wide variety of targets including ions [7,8], small molecules [9], proteins [10], and whole cells [11]. Graphene oxide-assisted SELEX (GO-SELEX) is an immobilization-free method that can be easily implemented to enrich the aptamer pool [12,13]. Graphene oxide presents the unique characteristic of adsorbing ssDNA on its surface due to hydrophobic and π–π stacking interactions between the nucleotide bases and GO, while repelling the negatively charged phosphate backbone of dsDNA [14].

In this study, we describe efforts in identifying single-stranded DNA aptamers capable of binding to a specific biomarker that uniquely signals the presence of Rs. MSA is considered an ideal target due to its uniqueness to Rs and high abundance in tissue during infection. It is a surface protein (over 50% of the surface protein content of Rs), immunodominant, and has no known homologues in GenBank [15]. The full-length MSA gene of the Rs bacterium as well as several truncations were successfully cloned to generate large quantities of the protein. Specifically, the study described here involved exploratory work to identify aptamers that bind to the MSA-R2c construct which contains an IPT (Immunoglobulin-like fold, Plexins, Transcription factors) putative DNA-binding domain and therefore is believed to be an effective binding site for ssDNA aptamers. We believe this is the first study reporting an aptamer binding to a surface protein of the Rs bacterium and could be important in the future development of aptamer-based diagnostic tools for rapid detection of Rs.

## 2. Results and Discussion

### 2.1. MSA Protein and Constructs Generation

p57 or major soluble antigen (MSA) is a glycine-rich, 558 amino acid cell surface protein containing two copies of an 81-residue direct repeat, which are 60% homologous, and five copies of a second 25 residue imperfect repeat [16]. MSA is the best-characterized virulence factor in Rs, although its role in the disease pathogenesis is unclear [2]. One function of MSA may be to mediate adherence of the bacteria to host tissues. MSA has been reported to form a fimbrial structure on the bacterial cell surface and has a number of other characteristics of bacterial adhesins including hemagglutinating activity, an acidic isoelectric point, and hydrophobic amino acid composition [16]. During an infection, MSA and its breakdown products are the principal constituents of the extracellular protein (ECP) fraction released by Rs into fish tissues and body fluids and the surrounding environment [17]. Because MSA is currently used as the antigen for detection of Rs by ELISA, it is considered a promising target for aptamer screening due to its uniqueness and high abundance. Analysis of the amino acid sequence shows the presence of a putative immunoglobulin-like domain, named after its presence in plexins and transcription factors (IPT) in the 239–302 amino acids region. Based on this analysis three regions were identified in the MSA gene and these regions were further divided to express six constructs, MSA-R1a (27–155), MSA-R2a (172–356), MSA-R2b (172–333), MSA-R2c (228–331), MSA-R3a (357–558) and MSA-R3b (357–483), in addition to the full length MSA (Figure 1).

For this study the MSA-R2c protein construct was chosen as the first sub construct to explore for aptamer binding. The construct was chosen due to its smaller size and high solubility in water while retaining the range of amino acids associated with the IPT domain. The construct was expressed in Escherichia coli BL21* cells and purified using a GST affinity tag and ion exchange chromatography. The size of 37 kDa and purity was confirmed by SDS-PAGE (Appendix A).

### 2.2. Isolation of DNA Aptamer

The label-free graphene oxide-assisted SELEX (GO-SELEX) process was chosen to isolate aptamers with affinity to the MSA-R2c protein (Figure 2).

The efficiency of GO-SELEX is due to the strong adsorption through electrostatic and π–π stacking interactions of ssDNA on the graphene oxide surface [14,18,19]. After optimizing the composition of binding buffer and the ssDNA:GO ratio [20], the GO-SELEX process was initiated by incubating the ssDNA random library and MSA-R2c protein in a 1:2 molar ratio. After separating unbound ssDNA by addition of GO and centrifugation, the recovered ssDNA bound to the MSA-R2c protein was amplified and labeled using a 5’-biotin modified primer. The optimal number of amplification steps was determined as previously described [21]. Optimization of PCR cycles is an important step in the GO-SELEX process to avoid overamplification and it was conducted for each SELEX round (Appendix A).

Streptavidin-coated agarose beads were utilized to separate ssDNA from the dsDNA generated during the PCR amplification. The eluted ssDNA was carried to the next selection round. The enrichment of target-specific aptamers was monitored at each round by calculating the recovery rate as the ratio of the amount of ssDNA in the enriched pool to the amount of ssDNA added at the beginning of the cycle (Figure 3). An increasing amount of ssDNA was observed up to round 4, after which the amount of eluted ssDNA remained fairly constant. A counter-selection step was introduced between round 5 and 6 to eliminate ssDNA that could potentially bind to the GST fusion protein. Presence of ssDNA after round 6 suggests enrichment of the pool with aptamers sequences specific for the MSA-R2c protein. After round 7, no significant increase in the pool enrichment was measured. The selection process was ended and the aptamer pool was cloned and sequenced.

### 2.3. Analysis of Aptamer Structures

Several groups of clones were analyzed and six different aptamer sequences were identified (Table 1) and their secondary structures were predicted by the Mfold software [22].

Minimum free energy structures of the aptamers predicted by the Mfold software are presented in Appendix A (binding buffer ionic conditions at 37 °C). All selected aptamers display similar structure patterns with the formation of stem loops in the random region. Several conserved motifs were also identified. Most frequently, a motif of TATTA was observed (5 aptamers), followed in frequency by GTT(T)A (4 aptamers) and TAAT motifs (3 aptamers each). Aptamer-4 had the highest frequency of occurrence in six different clones and the highest predicted folding stability with a minimum free energy of −7.23 Kcal/mol. To investigate the role played by the primers in the structure and stability of the full-length aptamer, a 40-mer truncated sequence was studied by removing the primer sequences at each end. However, the stability of the structure predicted by Mfold was dramatically decreased to a ∆G of −0.59 Kcal/mol indicating that the primer sequences play a role in stabilizing the aptamer structure. The full-length sequence of Aptamer-4 was therefore synthesized and analyzed for its binding affinity to MSA-R2c.

### 2.4. Binding Affinity and Specificity Test

Binding reactions were performed using a modified immuno-qPCR method. In immuno-PCR, a linker molecule with bispecific binding affinity for DNA and antibodies is used to attach a DNA molecule (marker) specifically to an antigen–antibody complex, resulting in the formation of a specific antigen–antibody–DNA conjugate. The attached marker DNA can be amplified by PCR with the appropriate primers. The presence of specific PCR products demonstrates that marker DNA molecules are attached specifically to antigen–antibody complexes, which indicates the presence of antigen [23]. In our approach, antibodies commonly used in immuno-PCR were replaced by the aptamer under study. qPCR is a simple, sensitive, and quantitative technique and has been used for the absolute determination of cell-bound aptamers during cell-SELEX [24] and for analysis of binding characteristics, including binding affinity (*K_d_*) and binding capacity (B_max_) of DNA-based aptamers for E. coli O157:H7 [25] and poly(C)-binding protein 2 (PCBP-2) [26].

A constant amount of MSA-R2c protein (125 pmol) was immobilized in multiple wells of an ELISA microplate and varying concentrations of Aptamer-4 were added and incubated according to the same conditions used in the SELEX protocol. In earlier experiments the free aptamer concentration was varied between 0.0205 nM and 175 nM and eventually limited between 0.0205 nM and 40 nM since saturation could be observed at lower concentrations. To release the bound aptamers from the MSA-R2c protein, the plate was heated at 95 °C for 10 min and the recovered aptamer was quantified by qPCR.

To quantitatively determine the concentration of aptamer for the binding assay, a calibration curve was first developed to verify a linear correlation between the cycle threshold (Ct) and the logarithm of the aptamer concentration with standards ranging from 0.005 nM to 2.04 nM. The standard curve showed excellent correlation with a R^2^ value of 0.9937 (Appendix A).

Subsequently, aptamer concentrations from the binding assay were estimated using the calibration equation. Figure 4 shows the binding curve obtained by plotting the fraction of ssDNA aptamer bound to the MSA-R2c protein as a function of free aptamer concentration. Bound aptamer concentrations were estimated by qPCR using the calibration equation described in Appendix A. The dissociation constant *K_d_* was calculated using a non-linear regression analysis and was found to be 3.0 ± 0.6 nM with R^2^ value of 0.99. No significant binding was measured when repeating the assay using random sequences as control. The *K_d_* value in the low nanomolar range indicates that Aptamer-4 has a high binding affinity for the MSA-R2c protein.

The specificity of Aptamer-4 was tested by evaluating its binding to bovine serum albumin (BSA). BSA is commonly used as a control protein to test the specificity of aptamers [26]. The binding assay was repeated using BSA as the target protein and same increasing concentrations of Aptamer-4. No detectable binding was observed between BSA and Aptamer-4, indicating a high degree of specificity of Aptamer-4 toward its target MSA-R2c protein.

## 3. Materials and Methods

### 3.1. Reagents and Instrumentation

The initial ssDNA library and primers (HPLC-purified), Prime Time Gene Expression Master Mix and LNA probe for qPCR were obtained from Integrated DNA Technologies (Coralville, IA, USA). Graphene oxide, pGEX6P-1 vector, glutathione-agarose beads were purchased from Sigma-Aldrich (St. Louis, MO, USA). Streptavidin-sepharose beads purchased from GE Healthcare Life Sciences (Pittsburgh, PA, USA) were used for separating biotinylated-ssDNA. SYBER gold nucleic acid gel stain was purchased from Thermo Scientific (Waltham, MA, USA). PCR reactions were performed using a PTC-100 Programmable Thermal Controller from MJ Research, Inc. (Waltham, MA, USA). qPCR reactions were run on a LightCycler 2.0 System (Roche Diagnostics Corporation, Indianapolis, IN, USA). Gels were visualized with a Gel Logic 112 Transilluminator, Carestream Health (Woodbridge, CT, USA) DNA concentrations were determined using a Thermo Scientific NanoDrop-2000 spectrophotometer.

### 3.2. Cloning, Expression and Purification of MSA Protein and Sub Constructs

The MSA gene was PCR amplified from the Rs genomic DNA obtained from the Northwest Fisheries Science Center, National Marine Fisheries Service (Seattle, WA, USA) and cloned into GST fusion pGEX6P-1 vector. The full length MSA (amino acids 27–558) and six constructs, MSA-R1a (27–155), MSA-R2a (172–356), MSA-R2b (172–333), MSA-R2c (228–331), MSA-R3a (357–558), and MSA-R3b (357–483) were cloned using BamHI and NotI restriction sites. Using BL21(DE3) chemically competent cells, protein expression was induced with 0.25 mM isopropyl β-D-1-thiogalactopyranoside (IPTG) to an OD_600_ ~ 0.5–1 at 37 °C for 3 h. Pellets from harvested cells were resuspended in a lysis buffer (1M NaCl, 25 mM Tris-HCl, pH 7.5, 10 mM β-mercaptoethanol, 0.1% Triton-X) for 30 min at 4 °C and lysed by sonication (3 × 30 s). After centrifugation at 17,000 rpm for 15 min, the supernatant containing the GST-MSA or constructs was incubated with 10 mL of gluthathione-agarose beads at 4 °C for 1 h. The purified protein was recovered by adding 15 mL of elution buffer (10 mM glutathione, 300 mM NaCl, 50 mM Tris-HCl, pH 8, 5 mM BME), rotating the mixture at 4 °C for 30 min, followed by additional 10 mL of elution buffer and incubation at 4 °C for 30 min. The protein was further purified using an anion exchange (Q) column (GE Healthcare), filtered through a 0.2 μ filter, and dialyzed overnight in the storage buffer and stored at −80 °C. The protein purity was verified by sodium dodecyl sulfate polyacrylamide gel electrophoresis (SDS-PAGE).

### 3.3. Random Oligonucleotide Library and Primers

An 80-nucleotide oligonucleotide ssDNA library consisting of a central 40 nucleotide randomized region flanked by two 20 nucleotide primer binding sites was used as the initial library (5′-ATGATACGGCGACCACCTAA-N_40_-CTTTCCCTACACGACGCTA-3′). The ssDNA library and subsequent aptamer pools were amplified by PCR with a forward primer (5′-ATGATACGGCGACCACCTAA-3′) and reverse primer (5′-TAGCGTCGTGTAGGGAAAGA-3′). A special biotinylated PCR primer (5′-biotin-TAGCGTCGTGTAGGGAAAGA-3′) was used for separating ssDNA from the double-stranded PCR product during the SELEX process. The amplification conditions were: 5 min at 94 °C, repeated cycles of 1 min at 94 °C, 1 min at 51 °C, 1 min s at 72 °C, followed by 5 min at 72 °C after the last cycle. The PCR cycles were optimized in each round of SELEX to obtain the intended 80 base pair products.

### 3.4. In Vitro Selection via GO-SELEX

Aptamer selection was performed in a manner similar to as described by Park [14]. Prior to being mixed with target MSA-R2c protein, the ssDNA library was dissolved in binding buffer (20 mM Tris-HCl at pH 7.4), heated at 94 °C for 5 min, and immediately cooled on ice for 15 min. In the first round, ssDNA (250 pmol) was incubated with MSA-R2c (500 pmol) in 500 μL of binding buffer for 2 h at 25 °C with tilting and rotation. Subsequently, 4.0 mg of GO was added to the mixture which was rotated at room temperature for 30 min. The mixture was centrifuged at 9500 rpm for 3 min, and the supernatant containing ssDNA bound to MSA-R2c was collected, while the non-binding ssDNA aptamers adsorbed onto GO were discarded. After recovering the ssDNA by ethanol precipitation, the potential aptamer pool was amplified by PCR using the biotinylated primers. The optimal number of PCR cycles was determined by starting with a low number of cycles (typically 6 or 8) and adding cycles at increments of 2 until the aptamer band was visualized by 2.5% agarose gel electrophoresis without smearing. The optimal number of cycles was then used for large-scale amplification to generate large quantities of PCR product. To recover ssDNA for the next cycle, the amplified pool was incubated with streptavidin-agarose beads and ssDNA was eluted with 200 mM NaOH. The recovered ssDNA was carried to the next SELEX cycle for further enrichment of the pool. The recovery rate was determined after each round to monitor the enrichment of target-specific sequences. The recovery rate was obtained by calculating the ratio of the amount of ssDNA in the enriched pool to the amount of ssDNA added at the beginning of the cycle. The amounts were determined by UV spectroscopy using a NanoDrop-2000 spectrophotometer. Counter GO−SELEX was performed starting from the sixth round to improve the specificity of the aptamers against MSA-R2c. The ssDNA recovered from the fifth round of GO–SELEX was incubated with GST (5 nmol) for 30 min. To obtain the aptamers that did not bind to GST, 4.0 mg of graphene oxide was added to the solution and incubated for 2 h. The solution was centrifuged at 9500 rpm for 5 min and the supernatant was discarded. The ssDNA bound to the GO surface was recovered by resuspending the GO-ssDNA in binding buffer and adding the target MSA-R2c protein (500 pmol) with incubation for two hours. The ssDNA bound to the target was recovered and amplified by PCR and ssDNA for the next GO-SELEX round was regenerated using streptavidin-agarose beads.

### 3.5. Cloning, Sequencing, and Structure Analysis of Selected Aptamers

The enriched aptamer pool from GO-SELEX round 7 was amplified by unmodified primers and cloned using the TOPO Cloning Vector (DNA TOPO TA cloning kit, Invitrogen, Waltham, MA, USA) and transformed in *E-coli*. The positive clones, identified by blue-white spot screening, were randomly selected and grown in TB Broth containing 100 µg/mL ampicillin, followed by plasmid DNA extraction by the QIA Prep Spin Miniprep kit (Qiagen, Germantown, MD, USA). The inserted aptamer DNA of each clone was sequenced and analyzed with a GenoneLab GeXP Capillary Array Sequencer, Beckman Coulter (Krefeld, Germany). The secondary structure of all candidates was predicted using the Mfold software [22].

### 3.6. Binding Assay Procedure

Binding assays were performed using a modified immunoPCR affinity binding assay [23,24,25,26]. In this procedure, wells of a Greiner Bio-OneTM 96-Well High Binding Standard ELISA microplate (ThermoFisher, Waltham, MA, USA) were coated with 50 μL of a 2.5 μM MSA-R2c protein and incubated at 4 °C for 24 h. The ELISA plate was washed twice with 20 mM Tris-HCl buffer pH 7.4. Furthermore, 1% BSA blocking buffer was added to each well and was incubated for 1 h at room temperature. The ELISA plate was washed with Tris-HCl buffer pH 7.4 and the supernatant was discarded. Increasing concentrations of Aptamer-4 (0.0205, 0.041, 0.205, 0.41, 2.05, 4.1, 10.25, 20.50, 30, 41 nM) were heat-denatured at 95 °C for 5 min and 50 μL of each individual concentration was added to each well coated with MSA-R2c protein (in triplicate). The experiment was repeated in three independent plates; thus, 9 wells were used for assaying each aptamer concentration. After incubating the plate at room temperature for 1 h, the wells were washed three times with Tris HCl buffer pH 7.4, and the supernatant was discarded. Total of 50 μL of ultra-pure water was added to each well and the ELISA plate was heat-denatured at 95 °C for 10 min to release the bound aptamer from the MSA-R2c protein. The supernatant was collected and placed on ice.

Quantification of the unbound aptamer was achieved with a qPCR assay. A master mix was prepared with Prime Time Gene Expression Master Mix, 0.149 µM LNA probe, associated primers, and aptamer from the previous step. A calibration curve was prepared with aptamer standards ranging from 0.005 nM to 2.04 nM. All samples were analyzed in three replicates using a LightCycler and aptamer concentrations resulting from the binding assay were calculated. PCR conditions used are as follows: an initial heat activation step at 94 °C for 5 min, 45 cycles of a three-step process at 94 °C for 10 s, 51 °C for 30 s, and 72 °C for 5 s, and a cooling step at 37 °C for 5 min. A control containing no MSA-R2c was included to verify non-specific binding of the aptamer to the ELISA microplate. Additionally, the binding assay was repeated with random 80-nucleotide ssDNA library sequences under the same conditions.

On the basis of these data a saturation curve was constructed and the dissociation constant *K_d_* was calculated by non-linear regression analysis using OriginPro 8 (OriginLab, Northampton, MA, USA) based on the following equation:(1)$$\Theta_{b}=\frac{\left[A\right]}{\left(K_{d}+\left[A\right]\right)}$$
where Θ_b_ is the fraction of aptamer bound to the protein, [*A*] is the concentration of free aptamer, and *K_d_* is the dissociation constant [27].

### 3.7. Specificity Test

To verify the specificity of Apatmer-4 to MSA-R2c, binding assays were conducted using bovine serum albumin. In one trial, 50 μL of a 2.5 μM solution of BSA was incubated with increasing concentrations of the aptamer (0.025–41 nM), followed by a subsequent experiment where the concentration of BSA was increased to 5 μM. Binding analysis was performed using q-PCR as described for the target MSA-R2c protein.

## 4. Conclusions

This study constitutes the first report of the isolation and characterization of an aptamer with great affinity to p57 or major soluble antigen (MSA). This protein is the principal antigen found on the surface of Rs, the bacterium responsible for bacterial kidney disease in salmonids and it is considered a specific biomarker that uniquely signals its presence in the environment and infected fish tissues. Analysis of MSA amino acid sequence shows the presence of a putative immunoglobulin-like domain, named after its presence in plexins and transcription factors (IPT) in the 239–302 amino acids region. Based on this analysis, three regions were identified in the MSA gene and these regions were further divided to express six constructs. MSA-R2c (aa 228–331) was chosen as initial exploratory target to investigate the protein’s ability to bind DNA. Through seven cycles of GO-SELEX, an aptamer (Aptamer-4) was identified that displays specificity and a strong binding affinity with a *K_d_* of 3.0 ± 0.6 nM toward the target MSA-R2c protein. Further studies will be necessary to assess the suitability of this aptamer for applications in sensor development when testing for the full-length protein obtained from the bacterium. Nonetheless, the results of this work are significant as they open up possibilities for future development of aptamer-based diagnostic platforms for rapid and onsite detection of Rs in water or in infected salmonids. Such tools could be applied as an early warning system for disease outbreaks and as an epidemiological tracking tool, significantly improving aquaculture management.

## Figures and Tables

**Figure 1 molecules-27-01853-f001:**
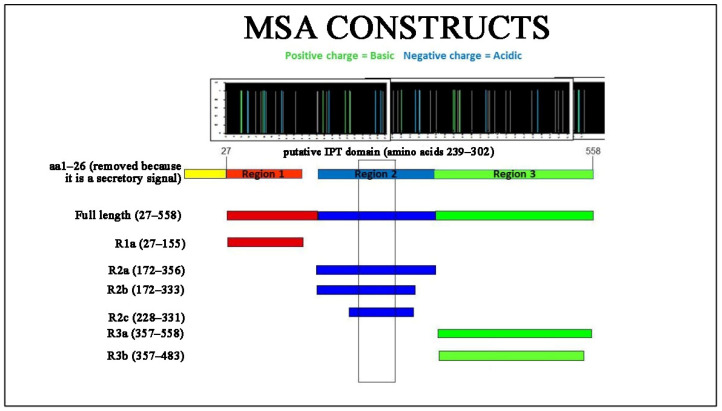
Six protein constructs were cloned and purified including full length MSA protein. Analysis of the amino acid sequence shows the presence of a putative immunoglobulin-like domain, named after its presence in plexins and transcription factors (IPT) in Region 2 which was identified as optimal binding sites. MSA-R2c was chosen as the first sub construct to explore for aptamer targeting.

**Figure 2 molecules-27-01853-f002:**
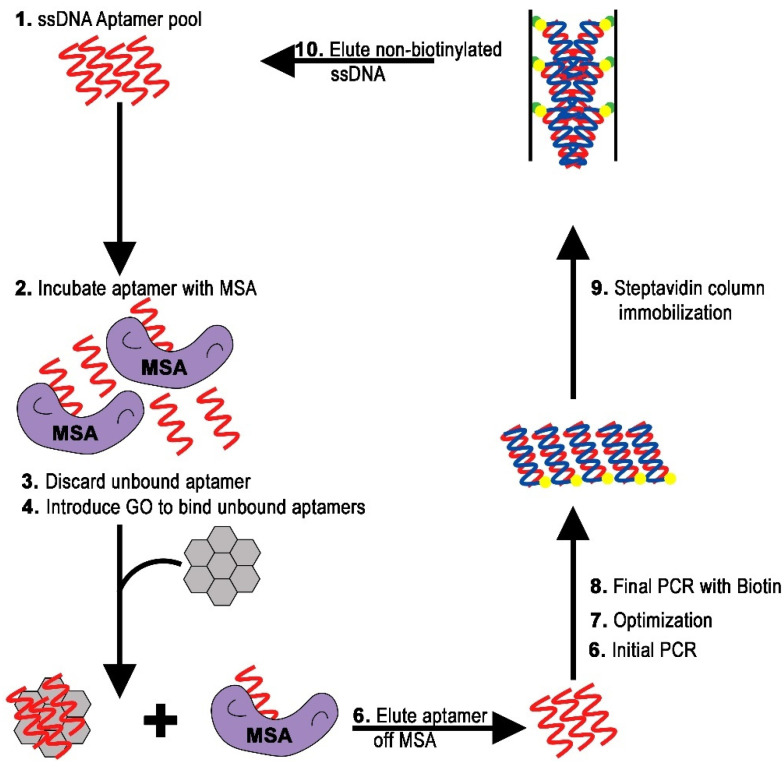
Aptamer production starts with a ssDNA random pool (top left) incubated with the target protein. Candidates with high binding potential proceed through several cycles of SELEX. Aptamers with low binding affinity for the target are separated and discarded via graphene oxide (bottom). Streptavidin-coated agarose beads are utilized to separate ssDNA from the dsDNA generated during PCR amplification. The eluted ssDNA is carried to the next selection round.

**Figure 3 molecules-27-01853-f003:**
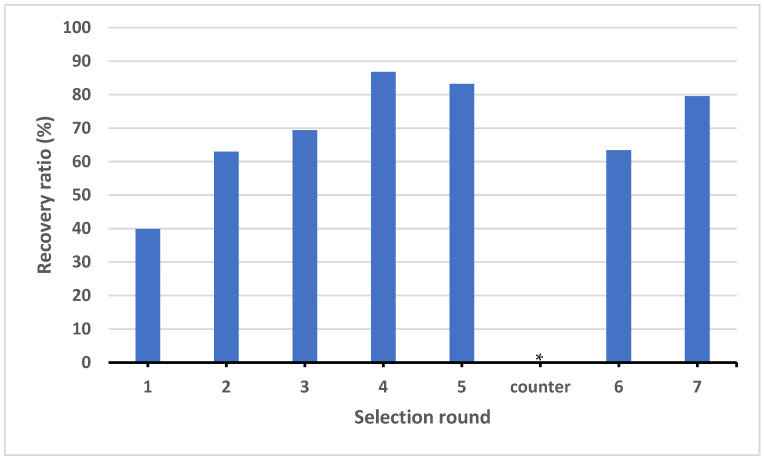
Enrichment of MSA-R2c specific aptamers throughout GO-SELEX. The graph represents the recovery rate after each selection cycle. The recovery rate was calculated as the ratio of the amount of ssDNA in the enriched pool to the amount of ssDNA added at the beginning of the cycle. The (*) represents a counter selection step introduced between cycles 5 and 6 to eliminate ssDNA that could potentially bind to the GST fusion protein.

**Figure 4 molecules-27-01853-f004:**
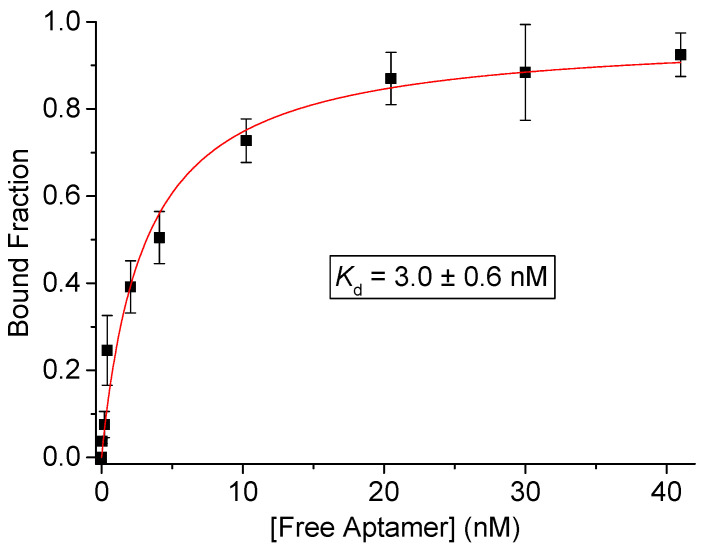
Binding of Aptamer-4 to MSA-R2c protein. The saturation curve was obtained by plotting the fraction of bound aptamer estimated by q-PCR as a function of free aptamer. The dissociation constant (*K_d_*) was estimated by non-linear regression analysis.

**Table 1 molecules-27-01853-t001:** Aptamers candidates identified from GO-SELEX.

Name	DNA Aptamer Sequence	Frequency
Aptamer-1	5′-ATGATACGGCGACCACCTAA**CGCACACAATATATTAAGGATCAAATTAATACTCGGTATA**TAGCGTCGTGTAGGGAAAGA-3′	1
Aptamer-2	5′-ATGATACGGCGACCACCTAA**ACATATGGAGTATTAGGGTTTACATAATTATGTATTTCTC**TAGCGTCGTGTAGGGAAAGA-3′	1
Aptamer-3	5′-ATGATACGGCGACCACCTAA**ATTACTACATTAATTAGTTGTATAATTGTAGTATGATGAT**TAGCGTCGTGTAGGGAAAGA-3′	2
Aptamer-4	5′-ATGATACGGCGACCACCTAA**CCATTTTCACTGTTTGTTTAATATATATAGTTTTATGGTA**TAGCGTCGTGTAGGGAAAGA-3′	6
Aptamer-5	5′-ATGATACGGCGACCACCTAA**AGTTATTTTTGTTATTATAACGGGTTACCTTTATAGACTAT**TAGCGTCGTGTAGGGAAAGA-3′	1
Aptamer-6	5′-ATGATACGGCGACCACCTAA**TTTATTAACTATTAACTGTTTAGGTGAGACTTTTCTGTTG**TAGCGTCGTGTAGGGAAAGA-3′	1

## Data Availability

The data presented in this study are available upon request from the corresponding author.

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
