# Peer review of "Isolation and Characterization of a ssDNA Aptamer against Major Soluble Antigen of Renibacterium salmoninarum"

_molecules, 2022, doi:10.3390/molecules27061853_

Round 1

Reviewer 1 Report

The manuscript by Layman1 et al. titled “Isolation and characterization of a ssDNA aptamer against Ma- 2 jor Soluble Antigen of Renibacterium salmoninarum” is describing the selection of ssDNA aptamers against Major Soluble Antigen (MSA) using SELEX as a combinatorial chemistry technique. The authors claimed the successful selection of several aptamers. the binding affinity showed that one of the aptamers can bind MSA protein with a Kd of 3.0 nM.  In my opinion, the isolation of aptamers against pathogens/pathogens proteins are appreciated making the current work gain adequate interest in the field. Overall, the manuscript is well written and provide important results. However, there are very important issues that prevent me from accepting this manuscript for publication in molecules, these issues need to be addressed from the authors:

  • This work needs more experiments to proof the binding of the selected aptamers to the bacteria, Renibacterium salmoninarum, why the authors did not perform a binding assay to proof that?
  • Although the Q-PCR is a valid technique to measure the fraction of the bound aptamers. However, gel mobility shifting assay was important to show the binding of the selected aptamer on gel. I can see that the authors did the PAGE analysis for the protein, so it was possible to perform gel mobility shifting assay with the aptamers.
  • In the binding experiment, did the authors used nonspecific sequence of library as control to eliminate the nonspecific binding?
  • Did the authors performed any stability assay to evaluate the degradation of aptamers?
  • The authors need to add the affinity binding equation to the materials and methods section.
  • The authors mentioned the use of 250 pmol in the first round. What is the diversity size of this quantity? Usually, we use 1013 to 1015 different sequences in the initial library.
  • I found it difficult to have the conclusion of using this aptamer to quantify the MAS protein without an experiment that proof that. How the authors can comment on this.

Reviewer 2 Report

In this paper, the authors proposed a novel aptamer with great affinity to p57 or Major Soluble Antigen (MSA) based on GO-SELEX. The isolation and identification of aptamers for potential diagnosis of Bacterial Kidney Disease (BKD) was proposed for the first time, which opens up possibilities for the development of aptamer-based diagnostics for rapid and onsite detection. Thus I recommend this work to be published on Molecules after revision, and some problems should be revised as follows:

  1.  In this manuscript, the structure of “Abstract” is not well organized and the research value of the work is not highlighted., hence, it is highly recommended to rewrite this part.
  2. For the study of the screened aptamers, only their structure and binding affinity were tested. This is a little simple, and perhaps more relevant experimental verifications can be added.
  3. In the Conclusion section, the authors briefly summarized this work, please consider to add more prospects for the future. What are the potential applications of this work?
  4. The authors may consider to improve the writing.Also, some figures are difficult to understand. For better understanding, the author needs to add extra description to the figure rather than simply figures only. Figure 3 is good example. What do the “counter”indicate?

Reviewer 3 Report

Review of the article “Isolation and characterization of a ssDNA aptamer against Major Soluble Antigen of Renibacterium salmoninarum”, by Layman et al.

This work presents interesting aptamer candidates to identify Renibacterium salmoninarum (Rs). The target used to develop the aptamers is a specific and abundant target, that means that the identification can be fast and effective as long as the specificity is verified.

This manuscript describes the development of a rapid detection method, the advantage that it proposes are the detection in field and with a short response time, perhaps without the need to sacrifice the animal to be able to recognize the presence of the pathogen.

The dissociation constant of the aptamer presented by the results is quite encouraging and this would represent a high potential of the aptamer, however the types of controls used during the binding test are not specified.

The main problem with the manuscript is the lack of any experiment to assess specificity. I consider necessary to add a negative test to be able to verify the specificity of the aptamer since during the experiments only a negative selection against the tag used in the different constructions of the ligand is mentioned, however, and even if the selected ligand is considered unique, it is very important to have negative controls of other proteins to be able to corroborate the specificity of the aptamer candidates. At least one other protein should be tested for affinity, ideally one that could be expected to cross-react in an undesirable fashion, such as a homologous protein in a non-pathogenic organism found in salmon or in water, or otherwise ~2 different proteins to reduce likelihood of good affinity, but poor specificity.

At lines 320-321 the diagnostic potential of Bacterial Kidney Disease (BKD) by means of the use of the characterized aptamer is mentioned; however the studies carried out were intended for laboratory constructions of antigen only, carrying out tests with real samples in order to be able to consider the potential of identification of the disease would be greatly beneficial for this, as well as analyze the conditions presented in the field environment. Nevertheless, it is only mentioned as “... for potential diagnosis...” and as such does not necessarily require field experiments.

From my opinion, the stationary phase of the aptamer enrichment is not clearly seen in the graph presented in Figure 3. There could be the possibility of further improving the enrichment with some more cycles of SELEX, but ultimately, if the selected aptamer answers the needs for sensing, this could suffice.

The results presented are based on experiments carried out with Aptamer-4, were any other sequence tested? If yes, it would be interesting to compare other candidates (perhaps in supplementary materials).

I consider line 323 to contradict the assertion of the diagnostic potential of Bacterial Kidney Disease (BKD) mentioned in the same paragraph on the conclusions, as the direction in which the objective of the study is directed should be considered; if the identification of the bacterium is sought we are only talking about identifying its presence, however we don’t specify its concentration in the sample analyzed, if disease identification is considered then the study should expand its experiments in order to know the limits of the minimum concentrations capable of detecting in the samples (Limit of Detection, LOD), in order to check that the sensitivity of the identification matches the infective dose of the pathogen.

Note that there are errors/typos, for example in Table 1 "Apatmer".

Round 2

Reviewer 1 Report

The authors have correctly answered to all the questions raised by the reviewer and improved their manuscript as asked. There is a need to provide a proof showing the ability of the selected aptamer to bind targeted bacteria in native status and this can be a future work.

Author Response

We appreciate the constructive suggestions provided by this reviewer.  Future work in this project will address the ability of the aptamer to recognize the MSA protein in a natural environment. 

Reviewer 3 Report

The authors answer part of our comments, but a minimal assessment of specificity for the aptamer is required. It could be as specific to MSA as SDS, which interacts with any protein. Actually, this can be especially problematic given the fact that MSA has a cdd00102 domain, which is known for its DNA-binding capacity, thus increasing the risk of non-specific interactions.

As mentioned in our previous comments, a succinct specificity assessment could be satisfactory. Ideally with a similar protein, but if easily available proteins (like BSA and/or lysozymes) are used to ensure that the aptamer is specific towards MSA, and not just any protein, this could be acceptable.
